# Production of Biodegradable Polymer from Agro-Wastes in *Alcaligenes* sp. and *Pseudomonas* sp.

**DOI:** 10.3390/molecules26092443

**Published:** 2021-04-22

**Authors:** R. Z. Sayyed, S. S. Shaikh, S. J. Wani, Md Tabish Rehman, Mohammad F. Al Ajmi, Shafiul Haque, Hesham Ali El Enshasy

**Affiliations:** 1Department of Microbiology, PSGVP Mandal’s, Arts, Science and Commerce College, SHAHADA, Maharashtra 425 409, India; sohels7392@gmail.com (S.S.S.); sonalwani1990@gmail.com (S.J.W.); 2Department of Pharmacognosy, College of Pharmacy, King Saud University, Riyadh 11564, Saudi Arabia; mrehman@ksu.edu.sa (M.T.R.); malajmii@ksu.edu.sa (M.F.A.A.); 3Research and Scientific Studies Unit, College of Nursing and Allied Health Sciences, Jazan University, Jazan 45142, Saudi Arabia; shafiul.haque@hotmail.com; 4Institute of Bioproduct Development (IBD), Universiti Teknologi Malaysia (UTM), Skudai, Johor Bahru 81310, Malaysia; henshasy@ibd.utm.my; 5City of Scientific Research and Technology Applications, New Burg Al-Arab, Alexandria 21934, Egypt

**Keywords:** biodegradable plastic, biodegradation, characterization, extraction, kinetics, production

## Abstract

The present study was aimed to evaluate the suitability of agro-wastes and crude vegetable oils for the cost-effective production of poly-β-hydroxybutyrate (PHB), to evaluate growth kinetics and PHB production in *Alcaligenes faecalis* RZS4 and *Pseudomonas* sp. RZS1 with these carbon substrates and to study the biodegradation of PHB accumulated by these cultures. *Alcaligenes faecalis* RZS4 and *Pseudomonas* sp. RZS1 accumulates higher amounts of PHB corn (79.90% of dry cell mass) and rice straw (66.22% of dry cell mass) medium respectively. The kinetic model suggests that the *Pseudomonas* sp. RZS1 follows the Monod model more closely than *A. faecalis* RZS4. Both the cultures degrade their PHB extract under the influence of PHB depolymerase. Corn waste and rice straw appear as the best and cost-effective substrates for the sustainable production of PHB from *Alcaligenes faecalis* RZS4 and *Pseudomonas* sp. RZS1. The biopolymer accumulated by these organisms is biodegradable in nature. The agro-wastes and crude vegetable oils are good and low-cost sources of nutrients for the growth and production of PHB and other metabolites. Their use would lower the production cost of PHB and the low-cost production will reduce the sailing price of PHB-based products. This would promote the large-scale commercialization and popularization of PHB as an ecofriendly bioplastic/biopolymer.

## 1. Introduction

The bountiful use of non-degradable synthetic polymers has created a frightening scenario for the environment [1]. Synthetic polymers possess numerous health and environmental hazardous at every stage of their existence. The environmental and health concerns brought poly-β-hydroxybutyrate (PHB) as sustainable and the best alternative to synthetic plastic and an ideal material for making biodegradable plastics [2,3]. Moreover, it is completely biodegradable in a natural environment after disposal [4,5]. Although PHB has been found as an ecofriendly biopolymer, the cost associated with carbon substrate (50% of the overall production) used in the fermentation, has been a major limiting factor in the commercialization of biodegradable polyesters [6,7].

PHB production from various oils like oil from spent coffee grounds by *Cupriavidus necator* H16 [8], soybean oil by *Ralstonia eutropha* [9], olive oil, corn oil by *Pseudomonas* sp. [10], and palm oil by *R. eutropha* [11,12] has been used for PHB production. Similarly, PHB production from low-cost and renewable carbon sources like industrial wastes such as malt, soya, sesame, molasses, bagasse, coconut pulp, and pharmaceutical waste have been reviewed and reported [13]. However, such substrates have resulted in less PHB yield and hence PHB production from inexpensive agricultural sources with high productivity may contribute significantly to lowering the production cost of PHB [14]. 

During nutrient starvation, organisms that accumulate PHB also degrade to get carbon and energy [4]. Such organisms having the potential of accumulating large amounts of PHB under carbon excess and nitrogen-deficient conditions and 90 capable of degrading this reserve food are of great commercial potential both in the production and degradation of biopolymer after its disposal [5]. The present study was aimed to evaluate agro-wastes and vegetable oils as potential substrates for higher yields of PHB by using *A. faecalis* RZS4 and *Pseudomonas* sp. RZS1. Since these strains were earlier reported to produce copious amounts of PHB in nitrogen-deficient medium (NDM) [15,16], they were used to check PHB production from agro-wastes and vegetable oils. The present research was carried out in the Microbiology Department of PSGVP Mandal’s College, Shahada, India from July 2018 to January 2020.

## 2. Results

### 2.1. Screening, Detection, Production, and Estimation of PHB

Growth of *A. faecalis* RZS4 and *Pseudomonas* sp. RZS1 appeared in the form of black-blue colored colonies. Addition of Sudan black B in NDM having growth of *A. faecalis* RZS4 and *Pseudomonas* sp. RZS1 resulted in the appearance of blue-black colonies. These colonies when smeared and stained with Sudan black B and observed under bright field microscopic revealed the presence of bright refractile granules (Figure 1).

### 2.2. Growth and PHB Accumulation as a Function of Time

*A. faecalis* RZS4 exhibited a lag phase of 6 h, exponential phase from 6 to 24 h, static phase from 24 to 30 h followed by decline phase. PHB accumulating began from 6 h and continued up to 48 h. However, an optimum amount of PHB accumulation occurred at 24 h (the late log phase) (56.92% of dry cell mass, i.e., 5.2 g/L) (Figure 2a). *Pseudomonas* sp. RZS1 exhibited a lag phase of 6 h, exponential phase from 6 to 30 h, static phase from 30 to 36 h followed by decline phase. PHB accumulation began from 12 h and continued up to 48 h. Maximum PHB accumulation occurred during the late log phase 30 h) (54.14% of dry cell mass, 4.2 g/L) (Figure 2b).

### 2.3. Kinetics of Biomass Growth and PHB Formation

Bacterial growth pattern exhibits 4 phases, i.e., lag phase—phase of acclimatization and no growth; log phase—phase of most active and rapid growth resulting in the doubling of the bacterial population; stationary phase—in which there is no growth, cell death occurs due to the depletion of nutrients, and the presence of toxic metabolites; and decline/death phase [17]. Bacterial growth is determined by the measurement of biomass, i.e., cell dry weight mass or optical density (OD) [18,19].

Kinetic growth models are useful as they assist engineers to design and control bioprocesses [20]. Monod’s model [21] is one of the earliest kinetic growth models that defined the specific growth rate. However, this model was not capable of describing specific growth rates in the presence of high (toxic) substrate concentration [22]. Gharibzahedi et al. [23] evaluated the kinetic growth models using the coefficient of determination (R²), i.e., the ratio of the explained variation to the total variation. R² is used to determine the efficiency of the model and its closeness to a value of 1 is an effective and practical measure of the validity of a model prediction [24]. The differential equations were solved using the ‘odeint’ solver of the ‘scipy’ module of Python 3.7 with the internal conditions of 0.1, 30, and 0 for the biomass, substrate, and product concentration respectively. The solution of the differential equation was fitted with the experimental data to calculate the kinetic parameters. The mathematical model fitted the experimental data of biomass growth and PHB production of both the micro-organism with good accuracy. The kinetic model of the *Pseudomonas* sp. Showed a very good fit for the biomass growth (R^2^ = 0.941, RMSE = 0.049) and PHB production (R^2^ = 0.992, RMSE = 0.135). The model predicted kinetic parameters of the *Pseudomonas* sp. Are µ_max_ = 2.03 × 10^−1^/h, K_s_ = 30 g/L, α = 8.63, β = 3.11 × 10^−14^, Y_x/s_ = 9.56 × 10^−2^ g biomass/g substrate, and Y_p/s_ = 1.74 × 10/g PHB/g substrate (Figure 3a). The *A. faecalis* RZS4 data also showed a good fit with the model for both biomass production (R^2^ = 0.783, RMSE = 0.079) and the PHB formation (R^2^ = 0.987, RMSE = 0.24) (Figure 3b). The kinetic parameters for the *A. faecalis* RZS4 predicted by the model are µ_max_ = 8.0 × 10^−2^ h, K_s_ = 3.08 g/L, α = 14.99, β = 1.88 × 10^−10^, Y_x/s_ = 8.64 × 10^−1^ g biomass/g substrate, and Y_p/s_ = 1.69 × 10^−1^ g PHB/g substrate. The kinetic model assumed that both the organisms follow the Monod model for growth. 

### 2.4. Extraction and Estimation of PHB and Cell Mass

Washing PHB containing cell mass with sodium hypochlorite (1% *v*/*v*) resulted in the disruption of the cell wall. Treatment with cell mass with a mixture of acetone: ethanol (1:1, *v*/*v*) resulted in the release of PHB granules. Further incubation of the digested cell mass with boiling chloroform produced crystalline powder of PHB. The amount of PHB of *A. faecalis* RZS4 and *Pseudomonas* sp. RZS1 was gravimetrically estimated as 79.90% of dry cell mass 66.22% of dry cell mass respectively. Spectrophotometric estimation of PHB produced by these organisms was 5.21 g/L and 4.32 g/L respectively.

### 2.5. Production of PHB from Agro-Waste

For sustainable production of PHB, various agro-wastes were checked for their suitability to support the production of PHB from these renewable and cheap carbon sources. *A. faecalis* RZS4 and *Pseudomonas* sp. RZS1 were separately grown at 30 °C for 48 h at 120 rpm in each NDM amended with 20 g/L of corn waste, rice straw, and wheat straw revealed that *A. faecalis* RZS4 yielded more PHB in medium containing corn waste as a carbon source (5.21 g/L, i.e., 541.46 µg of PHB/mg of cell mass). Whereas *Pseudomonas* sp. RZS1 produced maximum PHB in medium amended with rice straw as carbon source (4.31 g/L, i.e., 379.98 µg of PHB per mg of cell mass). *A. faecalis* RZS4 yielded more PHB but less growth with these agricultural wastes while *Pseudomonas* sp. RZS1 grew luxuriously but produced less PHB. 

### 2.6. Production of PHB from Crude Vegetable Oils

*A. faecalis* grown at 30 °C for 48 h at 120 rpm in NDM individually amended with 20 g/L of vegetable oil, yielded a higher level of PHB in NDM containing sesame oil (5.92 g/L, 66.21% of dry cell mass, and 368.98 μg of PHB/mg of cell mass) (Figure 4). *Pseudomonas* sp. RZS1 also produced higher PHB yield in NDM amended with sesame oil as a carbon source (5.48 g/L, i.e., 69.52% of dry cell mass equivalent to 333.33 μg of PHB/mg of cell mass) (Figure 5).

### 2.7. Characterization of PHB 

#### UV–Visible Spectrophotometry

*A. faecalis* RZS4 produced more amounts of PHB than *Pseudomonas* sp. RZS1and hence its PHB extract was subjected to UV–Visible spectrophotometric characterization. UV–Visible spectrophotometric studies of PHB extract of *A. faecalis* RZS4 revealed the presence of a single peak at 220 nm with an absorbance value of 3.436 (Figure 6). 

### 2.8. Assessment of Biodegradation of PHB 

The growth of *A. faecalis* RZS4 and *Pseudomonas* sp. RZS1 in MM amended with PHB extract as the only carbon source resulted in the formation of a zone of PHB hydrolysis (Figure 7). *A. faecalis* RZS4 produced a bigger zone of PHB hydrolysis vis-à-vis *Pseudomonas* sp. RZS1 (Figure 7).

PHB film added in NDM broth followed by inoculation with *A. faecalis* RZS4 and *Pseudomonas* sp. RZS1 exhibited visible changes (biodegradation) in the surface morphology, i.e., roughening of surface and formation of holes and distortion of PHB film (Figure 8a) as against no changes in the surface morphology of the film (Figure 8b). 

### 2.9. Assay of PHB Depolymerase Enzyme

PHB depolymerase activity in the case of *A. faecalis* RZS4 was higher (7.690 U) as compared to *Pseudomonas* sp. RZS1 (6.511 U). PHB depolymerase activity is the measure of the degree of PHB degradation.

## 3. Discussion

### 3.1. Screening and Detection of PHB Production

The appearance of blue-colored colonies following the addition of Sudan black B is due to the reaction of the stain with PHB granules. PHB granules possess an affinity for lipophilic stains such as Sudan black B. The intensity of the black-blue color of the colonies on NDM reflects the degree of PHB accumulation. This viable colony method has been regarded as the best and a rapid method for screening PHB accumulating cultures. It detects the presence of PHB granules in the cytoplasm of PHB accumulating bacteria [25,26,27]. The appearance of bright refractile granules in Sudan black B stained smears under a bright-field microscope has been reported as a confirmatory method of screening PHB-positive organisms [26]. 

### 3.2. Growth and PHB Accumulation as a Function of Time

A wide variety of bacteria under carbon-rich but nitrogen-deficient conditions accumulate PHB intracellularly as a source of stored food. Under carbon rich conditions, more and more PHB is accumulated [27]. Maximum growth and PHB accumulation by *A. faecalis* RZS4 and *Pseudomonas* sp. RZS1 in CRGM is attributed to the rich availability of carbon in this medium and deficiency of nitrogen in NDM. While a decline in PHB after 24 and 30 h incubation in *A. faecalis* RZS1 and *Pseudomonas* sp. RZS1 respectively is due to the exhaustion of available nutrients. Under nutrient deficiency organism mobilizes the PHB storage [26]. Bacteria capable of accumulating PHB have been reported to mobilize intracellular PHB under the influence of PHB depolymerase [28]. The enzyme PHB depolymerase degrades PHB into simpler carbon and nitrogen source. Accumulation of optimum amount of PHB during the late log phase has been reported in *A. faecalis* and *Microbacterium* sp. [15,24]. Matavulj and Molitoris [29] reported maximum PHB accumulation in *Agrobacterium radiobacter* during the stationary growth phase (96 h) of the organism under nitrogen-deficient conditions. Yüksekdağ et al. [30] observed PHB accumulation in *Bacillus subtilis* 25 and *Bacillus megaterium* 12 strains in nutrient broth. They reported the best PHB production yields of 18.03%, 14.79% after 45 h and 48 h in *B. subtilis* 25 and *B. megaterium* 12 respectively. They further observed the reduction in PHB production after 48 h, i.e., decline phase. Singh et al. [31] reported maximum PHB accumulation (5.191 g/L) during 72 h growth (stationary phase) of *Bacillus subtilis* NG220. They observed a significant decrease in PHB production after 72 h. 

### 3.3. Kinetics of Biomass Growth and PHB Formation

A very good fit of the kinetic data of growth and PHB production of *Pseudomonas* RZS1 is due to the reason that this organism follows the Monad’s kinetic model more closely. A less fit of the experimental data of growth and PHB production of *A. faecalis* RZS 4 indicated that this organism does not follow Monad’s Model very closely as compared to the *Pseudomonas* RZS1. Akthiselvan and Madhumalti [32] used a kinetic model that describes microbial growth and PHB accumulation to predict the performance of batch fermentation of *Bacillus safensis* EBT1. They reported a good fit of experimental data with the predicted values. The specific growth rate and PHB accumulation values for the Monod model were 0.16 and 79.51 g/L respectively. Their kinetics indicated that PHB accumulation in *B. safensis* EBT1 is growth-associated. The amount of PHB accumulated was found to increase with the increase in cell mass (log phase) and decrease with the decrease in cell biomass (stationary/death phase). Tripathi and Srivastav [33] reported maximum PHB yield (2.20 g/L) and optimum biomass (3.42 g/L) after 48 h growth of *Alcaligenes* sp. Their kinetic model revealed mixed growth associated with PHB formation.

### 3.4. Extraction and Estimation of PHB and Cell Mass 

The inefficient recovery processes have been one of the major causes that hampered the commercialization of PHB for a wide range of applications [6]. Various methods used for recovery of PHB cause severe degradation of PHB and therefore affect PHB recovery yield [34]. Since PHB accumulation is an intracellular process, its extraction needs cell lysis, PHB accumulating cells are known to become fragile and therefore are easily lysed. Sodium hypochlorite digestion of non-PHB cell mass (NPCM) resulted in the lysis of cells without affecting the PHB. Extraction of PHB produced from *Alcaligenes* sp. RZS 4 with the solvent system consisting of a mixture of alcohol and acetone (1:1 *v*/*v*) proved to be a specific and efficient recovery method capable of specifically lysing the NPCM without affecting PHB. Rawte and Mavinkurve [35], Gangurde and Sayyed [36], and Gangurde et al. [37] reported the mixture of an equal ratio of acetone and ethanol as the best solvent for the extraction of PHB. The solvent system consisting of a 1:1 mixture of alcohol and acetone proved to be a specific and efficient recovery method capable of specifically lysing the NPCM without affecting PHB i *Alcaligenes* sp. RZS 4 and *Pseudomonas* sp. RZS1. Recovery yields obtained with alcohol and acetone (1:1 *v*/*v*) were higher than earlier reported yields, 0.6 g/L (by the chloroform extraction method) [21] and 1.1 g/L (by the dispersion method) [38]. 

### 3.5. Production of PHB from Agro-Waste and Vegetable Oils

Production of PHB in NDM amended with corn waste, rice straw, and wheat straw by both cultures indicated the potential of the organisms to utilize these substrates as carbon sources. Gowda and Shivakumar [39] reported the production of poly hydroxyl alkanoates (PHA) by *Bacillus thuringiensis* IAM 12077 grown in various agro-wastes such as rice husk, wheat bran, bagasse, and straw as low-cost carbon substrates. Several studies recorded PHB accumulation in a wide variety of bacteria grown on agro-wastes [40,41,42]. Agro-wastes have been suggested as cheaper and renewable carbon substrates for the economical production of PHB. These wastes are generated abundantly and serve as rich sources of carbohydrates. Kulkarni et al. [43] reported the maximum accumulation of PHB in *Halomonas campisalis* MCM B-1027 with bagasse. Getachew and Woldesenbet [44] found optimum PHB accumulation in *Bacillus* sp. grown on cane bagasse, corn cob, and banana peel. Alsheherei [45] reported the production of PHB from fruit peel by using different species of *Bacillus* and found a good yield of PHB (47.61%). 

The synthesis of PHB using vegetable oils offers the possibility of generating a wide variety of free fatty acids and thus the chances of producing a wide variety of PHB having different copolymer constituents that may have great commercial potential. 

Vegetable oils are known to provide essential moieties of fatty acids that are required as precursors in the biosynthesis of PHA. There are reports on the production of PHB from olive oil, corn oil [10], and palm oil [12]. Arun et al. [46] reported PHB production in sesame oil by *Alcaligenes eutrophus*. Song et al. [47] reported production of the optimum amount of intracellular PHA in *Pseudomonas* sp. strain DR2 grown in NDM containing vegetable oil as the sole carbon source. Although various oils have been used for PHB production, however, the PHB yield has been limited to 37–40*%* [30]. Thakor et al. [48] reported the accumulation of medium chain length PHA in a medium that contained olive oil and sesame oil. More growth of *Pseudomonas* sp. in vegetable oils may be due to its ability to degrade oils and fats.

The suboptimal yield of PHB from agro-waste by both isolates reflected their inability to produce copious amounts of PHB vis-à-vis PHB yield obtained from vegetable oils. More PHB productivity from vegetable oils may be because of the fatty acids contents of the oils. Fatty acids such as erucic acid, oleic acid, palmitic acid, palmitoleic acid, stearic acid, and linoleic acid found in mustard and sesame oil are the precursors in the biosynthetic pathway of PHB [38,49]. Higher PHB yield by *A. faecalis* RZS4 and *Pseudomonas* sp. RZS1 from sesame and mustard oil indicated the diverse metabolic potential of these isolates.

### 3.6. Characterization of PHB 

UV–Visible Spectrophotometry

The presence of a single peak at 220 nm with an absorbance value of 3.436 corresponds to the peak and absorbance of PHB [30]. PHB and copolymers are also known to absorb between 230 and 254 nm [38,48]. The presence of characteristic absorption maxima indicated the presence of a single type of PHA. 

### 3.7. Assessment of Biodegradation of PHB 

To be an ecofriendly biopolymer, the PHB produced by the isolates should be biodegradable. The appearance of a zone of PHB hydrolysis in MM medium containing PHB as the only source of carbon indicated the ability of these cultures to degrade their PHB. No zone of hydrolysis on MM medium without PHB is due to the absence of growth due to a lack of carbon source (PHB). More hydrolysis of PHB by *A. faecalis* RZS4 indicated more biodegradation potential of PHB as compared to *Pseudomonas* sp. RZS1. This may be due to the more PHB depolymerase activity (7.690 U) in *A. faecalis* RZS4 as compared to *Pseudomonas* sp. RZS1 (6.511 U). Gangurde et al. [37] reported the production of higher amounts of PHB in *A. faecalis* and it is known that the PHB depolymerase activity is proportional to the amount of PHB accumulated by the cell. Therefore the higher PHB depolymerase activity is because of the accumulation of more amounts of intracellular PHB by *A. faecalis* RZS1.

Roughening of surface, formation of holes, and distortion of PHB film in NDM medium separately inoculated by *A. faecalis* RZS4 and *Pseudomonas* sp. RZS1 indicates the biodegradation of polymer while no changes in the surface morphology of the film PHB film in uninoculated medium (NDM+PHB film) rule out the possibility of autolysis or auto-oxidation of PHB [37].

The ability of organisms to hydrolyze PHB reflects their potential to produce PHB depolymerase. Mergaert et al. [50] observed PHB degradation by the clear zone method in 295 different soil isolates. Elbanna et al. [51] reported a similar observation on PHB degradation in *Schlegelella thermodepolymerans* and *Pseudomonas indica* K2. Sayyed and Gangurde [36] reported PHB accumulation in *Pseudomonas* sp. under aerobic and anaerobic conditions. Sayyed et al. [15,16] reported biodegradation of PHB by a *Stenotrophomomas* sp. and *Microbacterium* sp.

### 3.8. Assay of PHB Depolymerase Enzyme

The enzyme assay is based on the fact that PHB extract exposed to PHB depolymerase (supernatant) under the controlled buffered system, causes degradation of PHB polymer into monomers. Sayyed et al. [15] reported 6.675 U/mg/mL PHB depolymerase in *Microbacterium paraoxydans* RZS6. After 48 h (log phase) of incubation, RZS6 produced 6.675 U of PHB depolymerase with 0.247 mg/mL protein content in 2 days at 30 °C. Gowda and Srividya [19] reported the production of 4 U of extracellular PHB depolymerase with a protein content of 0.05 mg/mL from *Penicillium expansum*. After 4 days of incubation at 30° C at 120 rpm in MSM, *Stenotrophomonas* sp. RZS7 yielded 0.721 U/mL PHB depolymerase. The yield of PHB depolymerase in *A. faecalis* RZS1 is higher than the PHB depolymerase activities reported in *M. paraoxydans* [5], *Penicillium expansum* [52], and *Penicillium* sp. DS9701-D2 [44,53]. Sayyed and Chincholkar [37] reported the production of higher amounts of PHB in *A. faecalis* and it is known that the PHB depolymerase activity is proportional to the amount of PHB accumulated by the cell [54,55]. Therefore the higher PHB depolymerase activity is because of the accumulation of more amounts of intracellular PHB by *A. faecalis* RZS1.

Agro-waste like corn waste, rice straw, and wheat straw, is the rich source of carbon but deficient in nitrogen and are the best substrates for PHB production, as PHB production requires carbon-rich but nitrogen-deficient conditions. The results of kinetic models demonstrated a good fit with the model for both biomass and PHB production in *Pseudomonas* sp. RZS1 and *A. faecalis* RZS4. However, *Pseudomonas* sp. RZS1 follows the Monod model more closely as compared to *A. faecalis* RZS4. This indicated the more suitability of *Pseudomonas* spp. for the production of PHB from vegetable oils. Production of PHB from vegetable oils offers the opportunity of producing a variety of PHB having different copolymer constituents that may great industrial applications. It also offers greater economic feasibility to enhance the commercial and cost-effective production of PHB [54,55]. 

## 4. Materials and Methods

### 4.1. Source of Culture

Two bacterial cultures namely *Alcaligenes faecalis* RZS4 and *Pseudomonas* sp. RZS1 used in this study were obtained from the culture repository of PSGVP Mandal’s Arts, Science, and Commerce College, Shahada, India. These cultures were previously isolated from the local dumping yard and identified [6].

### 4.2. Culture Media and Growth Conditions

PHB accumulation is a two-stage process; the first phase of cell growth is followed by the second phase of PHB accumulation. For the cell the carbon-rich growth medium (CRGM) contained (g/L) glucose, 20; (NH_4_)_2_SO_4_, 2.0; KH_2_PO_4_, 13.3; MgSO_4_7H_2_O, 12; citric acid, 1.7; and 1.0 mL of trace element solution; containing (mL/L) FeSO_4_ 7H_2_O; 10, ZnSO_4_7H_2_O, 2.25; CuSO_4_5H_2_O, 1.57; MnSO_4_5H_2_O, 0.5, CaCl_2_2H_2_O, 2.0; Na_2_B_4_O_7_10H_2_O, 0.23; (NH_4_)6Mo7O_24_, 0.1; and pH was set to 7.2 [54]. The log phase cultures (10^−5^ cells/mL) of *A. faecalis* RZS4 and *Pseudomonas* sp. RZS1 were separately grown in CRGM at 30 °C at 120 rpm for 24 h. After incubation, broths were centrifuged at 10,000 rpm for 10 min and the cell mass of these cultures were grown at 30 °C at 120 rpm for 48 h in the PHB accumulation nitrogen-deficient medium (NDM) containing (g/L), Na_2_HPO_4,_ 3.8; KH_2_PO_4_, 2.65; NH_4_Cl, 2.0; MgSO_4_, 0.2; fructose, 2.0; EDTA, 5.0; ZnSO_4_. 7H_2_O, 2.2; CaCl_2,_ 5.45; MnCl_2_. 6H_2_O, 5.06; H_3_BO_3,_ 0.05; FeSO_4_7H_2_O, 4.79; NH_4_Mo, 24.4; CoCl_2_.6H_2_O, 1.6; and CuSO_4_5H_2_O, 1.57 [54]. Following the incubation, broth were centrifuged at 10,000 rpm for 10 min and the cell mass was subjected to the extraction and estimation of PHB.

PHB accumulation occurs in two stages, during the first stage cell growth occurs in the CRGM followed by the second stage of PHB accumulation in NDM. For this purpose, 1% inoculum of *A. faecalis* RZS4 and *Pseudomonas* sp. RZS1 were individually grown in each CRGM at 30 °C at 120 rpm for 24 h followed by centrifugation of broth at 10,000 rpm for 15 min to separate cell biomass. The cell biomass was washed with sterile distilled water and grown at 30 °C at 120 rpm for 48 h in each NDM followed by centrifugation and qualitative and quantitative detection, extraction, and estimation of PHB [56].

### 4.3. Source of Agro-Wastes and Vegetable Oils 

Agro-wastes such as corn waste, rice straw, and wheat straw were obtained from local farms. These substrates were hydrolyzed with 1 N HCl. Crude vegetable oils such as mustard, corn, sesame, and soybean oils were procured from the local market. 

### 4.4. Screening for PHB Production

*A. faecalis* RZS4 and *Pseudomonas* sp. RZS1 were screened for PHB accumulation by a viable colony method employing the use of Sudan Black B on nutrient agar [17]. For this purpose, each culture was grown (spot inoculation) on nutrient agar at 30 °C for 24 h followed by flooding with Sudan Black B solution (0.02% in ethanol) for 30 min and observed for the appearance of dark black-blue colored colonies [6]. The excess stain from colonies was removed by washing with ethanol (96%).

### 4.5. Detection of PHB by Sudan Black B Staining

The intracellular accumulation of PHB in the cytoplasm of isolates was detected by Sudan black B staining of cells grown in NDM [55]. For this purpose, a smear of cells from NDM was stained with 0.3% (*w*/*v*) Sudan Black B solution for 15 min followed by destaining with alcohol (50% *v*/*v*) and counterstaining with safranin solution (0.5%, *w*/*v*). The slides were then washed; air-dried and observed for the appearance of black-colored PHB granules in a red-colored cell under a bright field microscope with a micro image projection system (Model MIPS, Olympus, Mumbai, India) under an oil immersion lens.

### 4.6. Growth and PHB Accumulation as a Function of Time

For checking the optimum growth and PHB production as a function of time, log phase culture (5 × 10^−5^ cells/mL) of *A. faecalis* RZS4 and *Pseudomonas* sp. RZS1 were grown in NDM at 30 °C for 48 h at 120 rpm. Samples removed after 6 h intervals were assayed for measurement of growth and PHB [56].

### 4.7. Kinetic Study of Growth and PHB Formation

A mathematical model of the growth kinetics and PHB production was developed. It was assumed that the micro-organisms follow Monod Growth kinetics for growth and Luedeking and Piret equation for product formation. A very good fit of the experimental data with the suggested model reflects that the organism follows the kinetic model. Fructose was taken as the rate-limiting substrate. The rate of formation of biomass and product and the rate of substrate consumption can be given by the following equations:(1)$$\frac{dx}{dt}=\frac{\mu_{max}x}{k_{S}+S}$$
(2)$$\frac{dP}{dt}=\;\alpha \frac{\mu_{max}S}{k_{S}+S}x+\;\beta x$$
(3)dSdt=−μmaxSYxS(kS+S)x−1YPS(αμmaxSkS+S+βx)
where x is biomass, µ_max_ is the maximum specific growth rate, k_s_ is half-velocity constant, S is the substrate concentration, P is the product concentration, α and β are the empirical constants, Y_P/S_ is the yield of product per unit mass of the substrate consumed and, Y_x/S_ is the biomass formed per unit mass of substrate consumed. 

These equations were simultaneously solved using the ‘odeint’ function of the ‘scipy’ module of Python 3.7. The values of the constants were predicted by fitting the experimental data of biomass growth and PHB formation to the equations.

### 4.8. Extraction and Estimation of PHB and Measurement of Growth

PHB is accumulated inside the cell, therefore; extraction of PHB requires the lysis of PHB cell accumulating cells. Such cells become fragile and are easily lysed. For extraction of PHB, the cell mass from NDM was lysed by using sodium hypochlorite at 37 °C for 1 h, the content was centrifuged at 10,000× rpm for 10 min, the precipitate was washed with distilled water and PHB from cell lysate was extracted with a mixture of acetone:ethanol (1:1). The extracted PHB was precipitated in chloroform. The precipitate was dried at room temperature to derive the PHB in powder form [12]. PHB extract dissolved in a small amount of chloroform was measured by the gravimetric and spectrophotometric method [56]. In the gravimetric method, the PHB content of chloroform extract was determined as a difference in the weight of the sample before drying and after drying to constant weight. It was expressed as mg % of dry cell mass or g/L of the medium.

In spectrophotometric methods, the PHB content of the extract was quantitatively estimated according to the method of Law and Sleepecky [56] that converts PHB into crotonic acid. For this purpose, PHB extract (10–100 μg) was prepared in chloroform; the chloroform was evaporated by gentle heating at 40 °C. Of concentrated sulfuric acid 10 mL was added to each tube and incubated at 100 °C for 10 min. The solution was cooled to 28 °C and the amount of PHB present in the sample and cell mass were measured at 254 nm and 620 nm, respectively [56].

### 4.9. Production of PHB from Agro-Waste

To evaluate the effect of agro-waste like corn waste, rice straw, and wheat straw on the production of PHB, these substrates were acid hydrolyzed with 1 N HCl and were individually added at 20 g/L as the sole carbon source in each NDM. Each NDM was separately grown with log phase culture of *A. faecalis* RZS4 and *Pseudomonas* sp. RZS1 at 30 °C for 48 h at 120 rpm. After the incubation, cell mass separated by centrifugation was measured, and PHB was extracted and estimated 56].

### 4.10. Production of PHB from Crude Vegetable Oils

Vegetable oils are a good source of a wide variety of free fatty acids and thus provide good substrates for the biosynthesis of a diverse variety of PHB. These PHB with different copolymer constituents may have great commercial potential. For this reason, PHB production from vegetable oil was undertaken in NDM. For this purpose 20 g/L, each of vegetable oils such as mustard, sesame, and soybean oil were separately added as a carbon source in each NDM after the inoculation and incubation, measurement of cell growth and extraction and estimation of PHB were performed according to Law and Sleepecky [56].

### 4.11. Characterization of PHB Extracts 

#### UV–Vis Spectrophotometry

For this purpose, PHB extract (5.0 mg) dissolved in 5.0 mL of chloroform was scanned between 190 and 1100 nm on a UV–Visible spectrophotometer and observed for the absorption maxima characteristic of PHB [57]. 

### 4.12. Assessment of Biodegradation of PHB 

#### 4.12.1. Plate Assay

Plate assay is a semiquantitative estimation method used to screen polymer degrading organisms and for measuring the biodegradability of a polymer. To assess the biodegradability of PHB extract produced by *A. faecalis* RZS4 and *Pseudomonas* sp. RZS1, the log phase cultures of both organisms were separately grown (5 × 10^−5^ cells/mL) on NDM containing PHB extract (as the carbon substrate) at 30 °C for 48 h and observed for the formation of a clear zone around the colony as an indication of utilization/biodegradation of PHB [15].

#### 4.12.2. Shake Flask Method

Biodegradation of PHB was performed on PHB film prepared from PHB extract. For this purpose, PHB powder (0.15 gm) dissolved in chloroform (50 mL) was added into two Petri-plates and incubated at 30 °C for 10 h. PHB film was obtained by evaporation of chloroform at 50 °C. PHB films were separately added as the sole carbon substrate into two NDM, one flask was grown with log phase culture of *A. faecalis* RZS4 and another flask was grown with *Pseudomonas* sp. RZS1 at 30 °C for 8 days under shaking (120 rpm), samples withdrawn after 24 h interval were analyzed for qualitative estimation (weight loss) and quantitative estimation according to the method of Law and Slepecky [56].

#### 4.12.3. Assay of PHB Depolymerase Enzyme

In the enzyme assay, the polymer is exposed to a pH-controlled system containing PHB depolymerase, which mobilizes PHB into an oligomer or monomer. For this purpose, two NDM individually grown with *A. faecalis* RZS4 and *Pseudomonas* sp. RZS1 at 30 °C for 8 days were centrifuged (10,000× rpm for 10 min) and the supernatant was assayed for PHB depolymerase activity. The reaction mixture comprising of 100 μg/mL PHB (substrate), 2 mM CaCl_2_, and 0.5 mL culture supernatant was incubated at 30° C for 10 min and a decrease in absorbance due to biodegradation of PHB was measured at 650 nm. One unit of PHB depolymerase was defined as the amount of enzyme that decreases the absorbance by 0.1/min [57]

### 4.13. Statistical Analysis

All the experiments were performed in five replicates and the average values of five replicates were statistically analyzed by a one-way analysis of variance (ANOVA) followed by a Turkey’s test. Mean values of *p* ≤ 0.05 were considered statistically significant [58].

## 5. Conclusions

PHB has been recognized as the best and sustainable alternative to the present-day used hazardous petrochemical-based plastic polymers. PHB has found a wide range of applications in food, agriculture, biomedical, etc. However, the high cost of production associated with the use of costly substrates has hampered the large-scale commercialization of PHB for a wide range of applications. Thus there is a need to find cheap raw materials. In this regard, the agro-wastes and crude oils can serve as the best and sustainable substrates for the cost-effective production of PHB. The use of microorganisms that can convert the agro-wastes into commercially valuable products such as PHB can be the ideal candidate for the industrial production of PHB. The PHB accumulating bacteria such as *A. faecalis* RZS4 and *Pseudomonas* sp. RZS1 accumulated PHB during their growth on agro-wastes and crude vegetable oils and reflected a good correlation between biomass and PHB accumulation. These cultures also exhibited the potential to mobilize the stored PHB degraded the and isolates that show good kinetics for biomass and PHB accumulation can serve as a good the suitable candidate for large-scale production of PHB. Their use would lower the production cost of PHB and the low-cost production will reduce the sailing price of PHB-based products. This would promote the large-scale commercialization and popularization of PHB as an ecofriendly bioplastic/biopolymer.

## Figures and Tables

**Figure 1 molecules-26-02443-f001:**
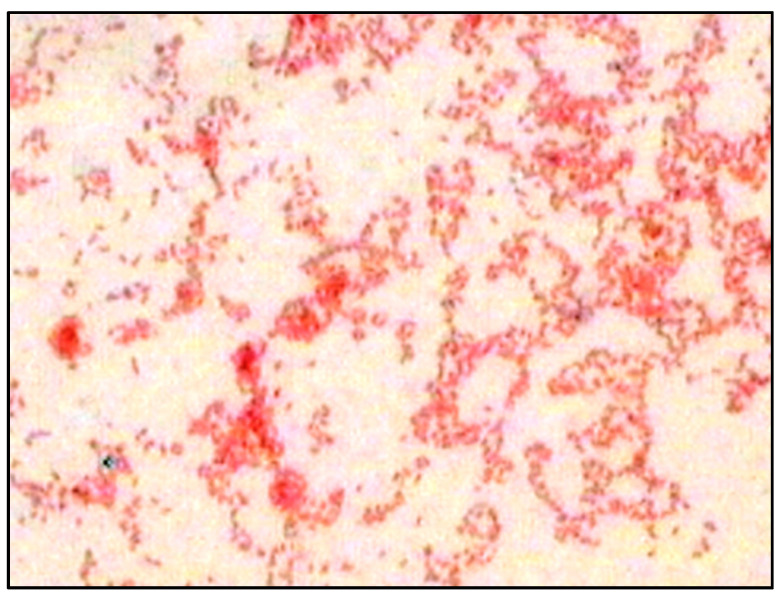
Microscopic observation of Sudan black B stained smear of *Alcaligenes* sp. RZS4 showing the presence of refractile PHB granules.

**Figure 2 molecules-26-02443-f002:**
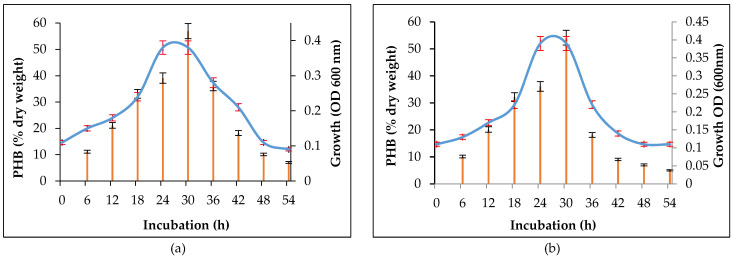
(**a**) Growth and PHB accumulation in *A. faecalis* RZS4) during growth in NDM broth at 30 °C for 48 h at 120 rpm. (**b**) Growth and PHB accumulation in *Pseudomonas* sp. RZS1 during growth in NDM broth at 30 °C for 48 h at 120 rpm. Samples were withdrawn at 6 h intervals and were subjected to the estimation of cell growth and PHB.

**Figure 3 molecules-26-02443-f003:**
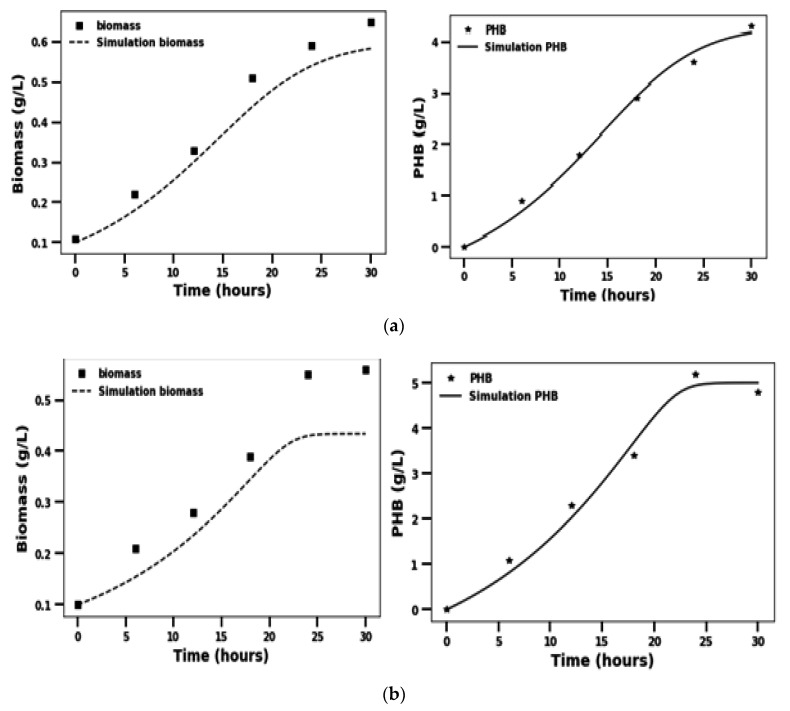
Kinetics of biomass growth and PHB formation by *Pseudomonas* sp. RZS4 (**a**) and *A. faecalis* RZS1 (**b**).

**Figure 4 molecules-26-02443-f004:**
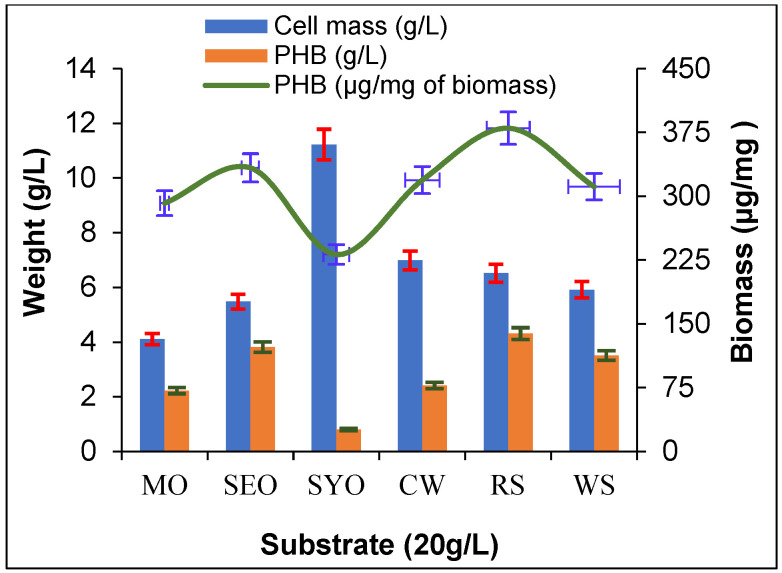
PHB accumulation in *A. faecalis* RZS4 during 24 h growth at 30 °C and 120 rpm in NDM amended with crude vegetable oil and agro-wastes (20 g/L) as carbon substrates.

**Figure 5 molecules-26-02443-f005:**
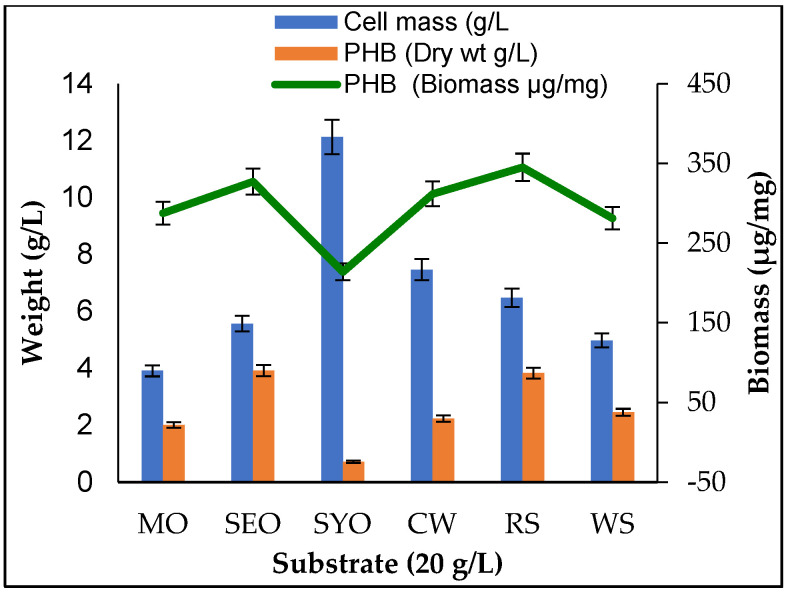
PHB accumulation in *Pseudomonas* sp. RZS 1 during 30 h growth at 30 °C and 120 rpm in NDM separately amended with crude vegetable oil and agro-wastes (20 g/L) as carbon substrates.

**Figure 6 molecules-26-02443-f006:**
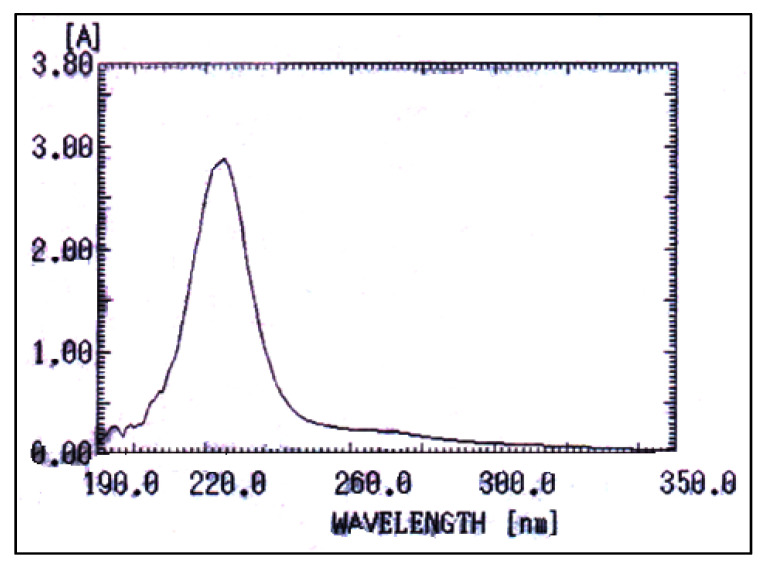
UV–Vis spectra of PHB extract of *A. faecalis* RZS4 obtained from NDM and extracted with acetone:alcohol.

**Figure 7 molecules-26-02443-f007:**
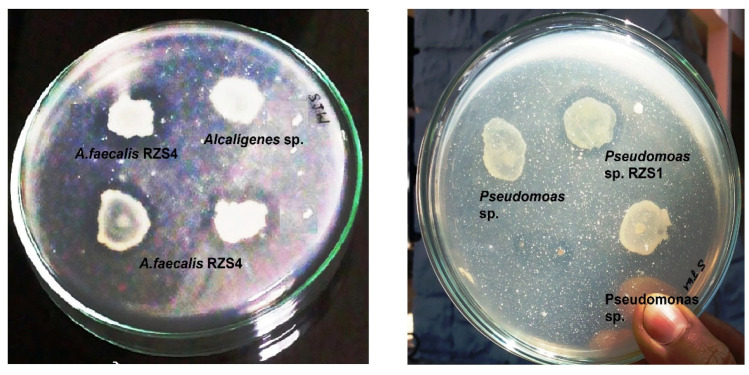
Biodegradation of PHB extracts of *A. faecalis* RZS4 and *Pseudomonas* sp. on MM having PHB extract (as the sole source of carbon grown) during 48 h incubation at 30 °C showing a zone of PHB hydrolysis around the colony

**Figure 8 molecules-26-02443-f008:**
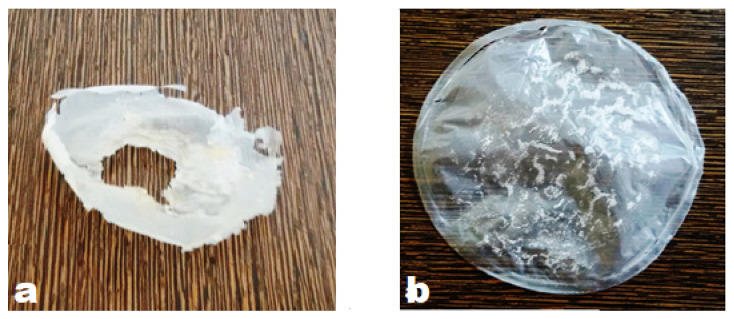
(**a**) Control preparation (uninoculated NDM +PHB film) did not show any change in the morphology of the film. (**b**) NDM inoculated *Pseudomonas* sp. RZS1 and *A. faecalis* RZS4 showing roughening of the surface morphology and formation of holes in PHB, i.e., indicative of the biodegradation of PHB film

## Data Availability

All the data is available in manuscript file.

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
