# Peer review of "Production of Biodegradable Polymer from Agro-Wastes in Alcaligenes sp. and Pseudomonas sp."

_molecules, 2021, doi:10.3390/molecules26092443_

Round 1

Reviewer 1 Report

The manuscript by these authors represents an interesting piece about the design and preparation (with particular care to the kinetic of growth) of bio polyesters. The manuscript was quite well written despite in some cases the acronym must be explained the first time is reported in the text and a plot (UV plot –figure 4) needs re-edit because it is in a non-presentable form at present. As regards the data, generally, R2 must be reported with four decimal places, furthermore, I would suggest to spent few words to introduce the readers to the kinetics model. Probably could be better to move the description of kinetic reported in the discussion just after the experimental. Finally, I would suggest strengthening some statements in the Introduction with specific and relevant references, seeing attached .pdf for this and other suggestions.

Author Response

Reviewer 1

Authors are thankful to the Reviewer for critical but excellent reviewing of the MSS. These comments have significantly improved the MSS

  • The manuscript by these authors represents an interesting piece about the design and preparation (with particular care to the kinetic of growth) of bio polyesters. The manuscript was quite well written despite in some cases the acronym must be explained the first time is reported in the text and

Authors’ response: All the acronyms are now explained at their first appearance.

  • A plot (UV plot –figure 4) needs re-edit because it is in a non-presentable form at present.

Authors’ response: Figure 4 is now re-edited

  • As regards the data, generally, R2must be reported with four decimal places,

Authors’ response: Agreed and the data of R2 is now presented as four decimal value. [Line No. 105,106,110].

  • I would suggest to spent few words to introduce the readers to the kinetics model. Probably could be better to move the description of kinetic reported in the discussion just after the experimental.

Authors’ response: Details of kinetic models have been added [Line No. 86-99]

  • Finally, I would suggest strengthening some statements in the Introduction with specific and relevant references, seeing attached .pdf for this and other suggestions.

Authors’ response: Suggested references have been added in the introduction section [Line 37-40].

Reviewer Comments as marked on pdf File

  • Line 34 : define abbreviation first time is used

Authors’ response: Acronym PHN was a typo [Line 29], it is now corrected as PHB that is already explained at Line 2

  • Line 44-46: I would suggest to strenghen this statement by citing references related with the persistence of polymers in the environment such as, Blanco, I. Lifetime Prediction of Polymers: To Bet, or Not to Bet—Is This the Question? Materials 2018, 11, 1383. https://doi.org/10.3390/ma11081383

Authors’ response: Agreed and the suggested reference has been added [Line No. 37].

  • Line 48: also in this case I would suggest to cite relevant references about the use of bio-polymers, D’Anna, A.; Arrigo, R.; Frache, A. PLA/PHB Blends: Biocompatibilizer Effects. Polymers 2019, 11, 1416. https://doi.org/10.3390/polym11091416

Siracusa, V.; Blanco, I. Bio-Polyethylene (Bio-PE), Bio-Polypropylene (Bio-PP) and Bio-Poly(ethylene terephthalate) (Bio-PET): Recent Developments in Bio-Based Polymers Analogous to Petroleum-Derived Ones for Packaging and Engineering Applications. Polymers 2020, 12, 1641. https://doi.org/10.3390/polym12081641

Authors’ response: Agreed and suggested references have been added [Line No. 40].

  • Line 92: I would suggest to spent few words to introduce the readers in the kinetics model.

Authors’ response: Two full paragraphs along with relevant citations have been added on the kinetic model [Line No. 86-99]

  • Line 99, 100, 103, 104: Generally for R2 four decimal places must be reported

Authors’ response: R2 is now given in four decimal value [Line No. 105,106,110].

  • Fig 1: it is not acceptable for publication, please edit this plot

Authors’ response: Figure 1 (Now Fig 2) is now edited

  • Line 206: probably it could be better the presentation of the model between the experimental and discussion sessions

Authors’ response: Agreed. Suggested revision has been made.

Reviewer 2 Report

The article entitled "Production of biodegradable polymer from agro-wastes in Alcaligenes sp. and Pseudomonas sp." addresses current problem of replacing synthetic polymers by the biobased. The research on alternative methods of PHB obtaining is a current issue, since the  cost-effectiveness needs to achieved. Although the topic of subject paper is interesting the paper needs major revision in all parts of the manuscript.

Major comments:

  1. Presented discussion for paragraphs 3.1-3.4 is poor and needs to be extended otherwise is more description of the results or methodology.
  2. The article is disorganized and I get the impression that some parts could be discussed together.
  3. Results of grow of both bacteria strains (Results, 2.1) was analyzed by optical light microscopy and those results are written in way that reader expecting to see pictures. Part 3.1 is not the discussion, is only an explanation of why this method was chosen (remark above).
  4. According to ref. [38] mixture of acetone and alcohol was used for washing of lipid granules not for extraction. For extraction of PHB boiling chloroform was used. Please add correct description on following sections: 2.4 (Results), 3.4 (Discussion), 2.9 (Materials and methods)
  5. Part Materials and methods and its paragraphs should be numbered as 4. In this part:
  • Part 2.2 and 2.6 should be marge – both are related to PHB production.
  • 3 – What was the product of substrates hydrolysis? Did Authors made some characterization after hydrolysis?
  • 9 – What was the unit of 1:1 aceton: ethanol ratio?
  • 11 – The statement “as described earlier” suggest that reference [37] refers to the authors' previous work.
  • 14 – Is the red highlight necessary?
  1. Please add axle descriptions on figure 3a and 3b. Authors should try to distinguished error bars (e.g. by the color) since the chart is difficult to read.
  2. The quality of Figure 4 is unacceptable.
  3. Figure 5 presents the same picture.
  4. Figure 6 pictures of the samples are mixed-up.
  5. Please uniform the shortcut “NDM”, “NDMM”, “MM” unless each refers to other composition.

Reviewer 2 Report

Major comments

Authors are thankful to the Reviewer for critical but excellent reviewing of the MSS. These comments have significantly improved the MSS

  • Presented discussion for paragraphs 3.1-3.4 is poor and needs to be extended otherwise is more description of the results or methodology.

Authors’ response: Discussion of 3.1 to 3.4 is now extended. Additional discussion on these aspects is now added [Line No.181-183, 185-186, 189-190, 192-194, 203-211, 213-225, 227-243].  

  • The article is disorganized and I get the impression that some parts could be discussed together.

Authors’ response : The article is now properly organized and some parts have been discussed together. Production of PHB from Agrowastes (3.5) and Production of PHB from vegetable oils is no discussed together [Line No. 244-279]

  • Results of grow of both bacteria strains (Results, 2.1) was analyzed by optical light microscopy and those results are written in way that reader expecting to see pictures.

Authors’ response: Figure of microscopic observation of Sudan black B stained smear is added [Figure 1].

  • Part 3.1 is not the discussion, is only an explanation of why this method was chosen (remark above).

Authors’ response: Agreed. This part is now revised and discussed in more details. [Line No.181-183, 185-186, 189-190]. 

  • According to ref. [38] mixture of acetone and alcohol was used for washing of lipid granules not for extraction. For extraction of PHB boiling chloroform was used. Please add correct description on following sections: 2.4 (Results), 3.4 (Discussion), 2.9 (Materials and methods)

Authors’ response : Agreed and the suggested corrections have been made in 2.4 [(Line No.120-123]  ), 3.4 (Line No. 227) and 2.9 (Line No.178-179)

  • Part Materials and methods and its paragraphs should be numbered as 4. In this part:

Authors’ response: Agreed and needful has been done. Materials and methods and its paragraphs are numbered as 5 as new heading Conclusion has been added at No. 4

Part 2.2 and 2.6 should be marge – both are related to PHB production.

Authors’ response: Agreed and 2.6 is now merged in 2.2 [Line No.354-374]. 

7) – What was the product of substrates hydrolysis? Did Authors made some characterization after hydrolysis?

Authors’ response: The present study was aimed only to observe the ability of organisms to hydrolysis PHB added in MM as the only source of carbon.

8) – What was the unit of 1:1 aceton: ethanol ratio?

Authors’ response: The unit of acetone and alcohol was v/v. [Line No. 122, 235, 242]

9) – The statement “as described earlier” suggest that reference [37] refers to the authors' previous work.

Authors’ response: This statement is now revised as according to the method of Law Sleepecky [Line No.431]

10) – Is the red highlight necessary?

Authors’ response: Not necessary. The red highlight is now removed Sleepecky [Line No.487-489]

  • Please add axle descriptions on figure 3a and 3b. Authors should try to distinguished error bars (e.g. by the color) since the chart is difficult to read.

Authors’ response: Figure 3a and b (Now Figure 4a and 4b) have been revised. Error bars have been distinguished by different colors

  • The quality of Figure 4 is unacceptable.

Authors’ response: The quality of Figure 4 (Now Fig 5) is improved with respect to its clarity and resolution

  • Figure 5 presents the same picture.

Authors’ response: Agreed. This Figure is now replaced

  • Figure 6 pictures of the samples are mixed-up.

Authors’ response: The pictures are now quite separated from each other

  • Please uniform the shortcut “NDM”, “NDMM”, “MM” unless each refers to another composition.

Authors’ response: Abbreviations are now uniform. NDM is used in all places.

Reviewer 3 Report

The use of petroleum-based polymers has provided quality and confort for the past few decades. These polymers are extremely persistent in the environment and none of the conventional techniques can degrade such polymers. A solution to this problem is the application of biodegradable polymers from different organic sources. For this reason the subject of this article is important.

I think this article should not be published because it has many imperfections:

  • I could not understand the innovation of this work, because there are several published works, where they use different types of bacteria and several inexpensive substrates
  • The kinetic study of growth and PHB formation needs to be further developed and explained
  • PHB property assessment is missing
  • Conclusions are missing

Author Response

Authors are thankful to the Reviewer for critical suggestion on the novelty of the work and for excellent Reviewing of the MSS. These comments have significantly improved the MSS

  • I could not understand the innovation of this work, because there are several published works, where they use different types of bacteria and several inexpensive substrates

Authors’ response: The novelty of the present work is mentioned in the Abstract [Line No 24-31], Discussion [Line No. 327-336] and Conclusion [Line No.338-346]. The present study deals with the screening of potent strains of Alcaligenes sp. and Pseduomonas sp. for higher accumulation of PHB. These cultures accumulated PHB during their growth on various agro-wastes and crude vegetable oil. Utilization of these wastes for the production of PHB is expected to give a sustainable solution for the management of such wastes. Higher production cost has been the major limiting factor in the production of PHB and its use for wide range of applications.

  • The kinetic study of growth and PHB formation needs to be further developed and explained

Authors’ response: Agreed and as suggested the kinetic study of growth and PHB formation is now be further developed and explained. [Line No.86-89]

  • PHB property assessment is missing

Authors’ response: The present study was aimed to screen potent PHB accumulation cultures and evaluate their ability to produce PHB from agro-wastes and crude vegetable oils as well as to access the their ability to mobilize the PHB storage under nutrient deficient conditions

  • Conclusions are missing

Authors’ response; Conclusion is now added [Line No.338-346].

Round 2

Reviewer 2 Report

Part 3.1 - repeated sentences related to lipophilic properties of stain used in the studies.

There are typos in section 3.3 e.g. line 226 in word "their", line 229 "biomass", please check carefully also Conclusion part.

Figure 7 - in my opinion still the description of pictures in the main text do not correspond with the figure description.

Author Response

Reviewer 2 Round 2 Report

Authors are thankful to the Reviewer for such an excellent and critical review of the MSS. The suggestions and comments helped in the significant improvement of the MSS

  • Part 3.1 - repeated sentences related to lipophilic properties of stain used in the studies.

Author Response : The repeated sentence related to lipophilic properties of stain used in the studies is now deleted from 3.1. [Line No. 189-190]

  • There are typos in section 3.3 e.g. line 226 in word "their", line 229 "biomass", please check carefully also Conclusion part.

Author Response :The typos in section 3.3 and other places in the manuscript have been corrected [Line No.24,27,31,36,39,47,48,50,62,63,65,67,69,88,96,100,114,121,158,178,182,183,185,191,192,197,203,209,220-222,225,228,230,232,234,239,,243,245,286,288,295,331,362,363,3,477,501,502].Conclusion part is revised in the light of results obtained [line No. 343-350]

  • Figure 7 - in my opinion still the description of pictures in the main text do not correspond with the figure description.

Author Response : Agreed. The description of the pictures (Fig. 7.) in the main text is now revised.